# Open Porous $\alpha + \beta$ Titanium Alloy by Liquid Metal Dealloying for Biomedical Applications

**Stefan Alexander Berger** [1,*] and **Ilya Vladimirovich Okulov** [2,3,4,*]

[1] Helmholtz-Zentrum Geesthacht, Institute of Materials Research, Division of Materials Mechanics, 21502 Geesthacht, Germany

[2] Faculty of Production Engineering, University of Bremen, Badgasteiner Str. 1, 28359 Bremen, Germany

[3] Leibniz Institute for Materials Engineering—IWT, Badgasteiner Str. 3, 28359 Bremen, Germany

[4] Institute of Natural Sciences and Mathematics, Ural Federal University, 620000 Ekaterinburg, Russia

[*] Correspondence: stefan.berger@hzg.de (S.A.B.); i.okulov@iwt.uni-bremen.de (I.V.O.);
Tel.: +49-(0)4152-872611 (S.A.B.); +49-(0)421-21851215 (I.V.O.)

**Abstract:** Open porous dendrite-reinforced TiMo alloy was synthesized by liquid metal dealloying of the precursor $Ti_{47.5}Mo_{2.5}Cu_{50}$ (at.%) alloy in liquid magnesium (Mg). The porous TiMo alloy consists of $\alpha$-titanium and $\beta$-titanium phases and possesses a complex microstructure. The microstructure consists of micrometer scale $\beta$-titanium dendrites surrounded by submicrometer scale $\alpha$-titanium ligaments. Due to the dendrite-reinforced microstructure, the porous TiMo alloy possesses relatively high yield strength value of up to 180 MPa combined with high deformability probed under compression loading. At the same time, the elastic modulus of the porous TiMo alloy (below 10 GPa) is in the range of that found for human bone. This mechanical behavior along with the open porous structure is attractive for biomedical applications and suggests opportunities for using the porous TiMo alloy in implant applications.

**Keywords:** dealloying; liquid metal dealloying; nanoporous; titanium alloy; biomedical material; mechanical behavior

## 1. Introduction

Titanium and titanium alloys are widely used as structural biomedical materials due to an excellent combination of mechanical and biological compatibilities [1,2]. This fact has stimulated an increasing interest in the development of novel titanium-based materials as well as improving the existing materials over the recent years [3–8]. One of the motivating factors for biomedical titanium alloy research was the so-called, "stress-shielding" effect, causing bone degeneration and implant loosening [1]. This effect is caused by the significant mismatch of stiffness between metallic material and bone. To overcome the "stress-shielding" problem, design strategies of several materials have been established including, for example, low modulus beta-titanium alloys [9–11], composite materials [12–14] and porous materials [8,15]. Recently, a series of low modulus porous titanium alloys possessing moderate strength [14,16,17] have been developed by liquid metal dealloying [18–21].

Liquid metal dealloying (LMD) is metallurgical method for synthesis of porous materials with the pore size ranging from several nanometers [19] to several micrometers [22,23]. LMD implies selective dissolution of one or more elements from a precursor alloy immersed in liquid metal. The typical liquid metal used for synthesis of porous titanium alloys is magnesium and magnesium alloys [5,24]. The design principles for precursor alloys are described elsewhere [18,19]. Since the discovery of LMD, it has been used to fabricate a wide range of porous metallic materials, including steels [22,25–30], titanium (Ti) alloys [14,16,17,24,31], zirconium alloys [5,16], tantalum (Ta) [32], niobium (Nb) [33,34],

cobalt-chromium (Co-Cr) [35], etc., as well as non-metallic porous materials such as silicon (Si) [36], carbon (C) [37–40], etc. The versatility of LMD can be emphasized through the synthesis of highly reactive materials such as meso/macroporous magnesium (Mg) [41] as well as compositionally complex alloys such as meso/macroporous high-entropy alloys (HEAs) [19,42]. Moreover, LMD has been used for the surface functionalization of biomedical titanium alloys such as Ti-6Al-4V and Ti-6Al-7Nb to improve their biological compatibility [23,43]. Finally, LMD has been used for synthesis of high-coercivity permanent magnets by laser powder bed fusion [44].

In this study, we developed open porous TiMo alloys reinforced by dendrites by liquid metal dealloying. Mo was selected as an alloying element due to several reasons: (i) Mo is immiscible with Cu [45], which leads to the formation of TiMo-rich dendrites in the precursor alloy; (ii) Mo is the beta stabilizing element in titanium alloys [46]. The microstructure and mechanical behavior of the porous TiMo alloy is stressed in detail.

## 2. Methods

The rod samples (1 mm in diameter) were fabricated from the $Ti_{47.5}Mo_{2.5}Cu_{50}$ (at.%) alloy via arc-melting and suction casting (MAM-1, Edmund Buehler GmbH, Bodelshausen, Germany) under the pure Ar atmosphere. The titanium-copper alloy for the rods was fabricated from the pure metals, namely titanium granules (99.99%), copper foil 1 mm (99.99%) and molybdenum wire (99.95%), supplied by Alfa Aesa GmbH. For homogenization, the alloy was remelted at least 20 times. The rods of 1 mm diameter were cut into 2 mm pieces by a wire saw (Model 3032, Well Diamantssäger, Mannheim, Germany) and these pieces were dealloyed in liquid Mg (magnesium granules with a purity of 99.98% supplied by Alfa Aesa GmbH, Kandel, Germany). Specifically, for the dealloying the sample pieces were put in a glassy carbon crucible together with Mg granules and heat treated in an IR-furnace (IRF 10, Behr, Düsseldorf, Germany) under the Ar gas flow. The precursor samples were dealloyed using several dealloying parameters, namely 800 °C for 10 min and 900 °C for 5 min. The temperature was chosen above the melting point of Mg to ensure the complete melting of the Mg granules. The dealloying time was selected to reach full dealloying of the samples and relatively small ligament sizes. During the dealloying, the Cu element was selective diffused from the master alloy into Mg melt, Ti and Mo atoms were rearranged into a bicontinuous structure and a nanocomposite consisting of Ti-rich and Mg-rich phases was formed. To obtain porous samples, the nanocomposite was chemically etched in 3 M nitric acid ($HNO_3$) for 24 h to remove the Mg-rich phase. The microstructure of the porous TiMo samples was investigated by means of scanning electron microscopy (SEM, Supra 55VP, Carl Zeiss AG, Oberkochen, Germany), X-ray diffractometry (XRD, D8 Advance, Bruker, Billerica, MA, USA) and nanotomography (Xradia Ultra 800, Carl Zeiss AG, Oberkochen, Germany). The reflections in the X-ray diffractograms were identified using X'Pert High Score software (Malvern Panalytical, Malvern, UK). The analysis of nanotomography data for ligament sizes was performed with the BoneJ tool in Fiji [47]. The chemical composition of the porous TiMo samples was executed by means of energy dispersive X-ray analysis (EDX, X-Max EDX system, Oxford Instruments, Abingdon, UK) in the SEM. The mechanical behavior of the cylindrical porous samples was tested in compression at room temperature and a strain rate of $10^{-4}\,s^{-1}$ using a universal testing device (Z010 TN, Zwick-Roell, Ulm, Germany). The strain was measured by a laser extensometer (LaserXtens, Zwick, Ulm, Germany). Three compression test samples were tested for each parameter set.

## 3. Results and Discussion

Figure 1 shows the diffraction patterns of the porous TiMo samples dealloyed at two different dealloying conditions. According to XRD analysis, the samples obtained using the different dealloying parameters both consist of α-Ti (hexagonal close-packed) and β-Ti (body centered cubic) phases. The Mo metal is the β-isomorphous alloying element for titanium, which stabilizes the β-Ti phase [46]. The synthesized α + β phase structure indicates that the concentration of Mo is below the critical one required to fully obtain the β-Ti alloy. The β-isomorphous alloying elements, such as Nb, β-eutectoid

and Fe, have been previously used to stabilize the β-Ti phase in the open porous Ti-based alloys obtained by liquid metal dealloying [5,17].

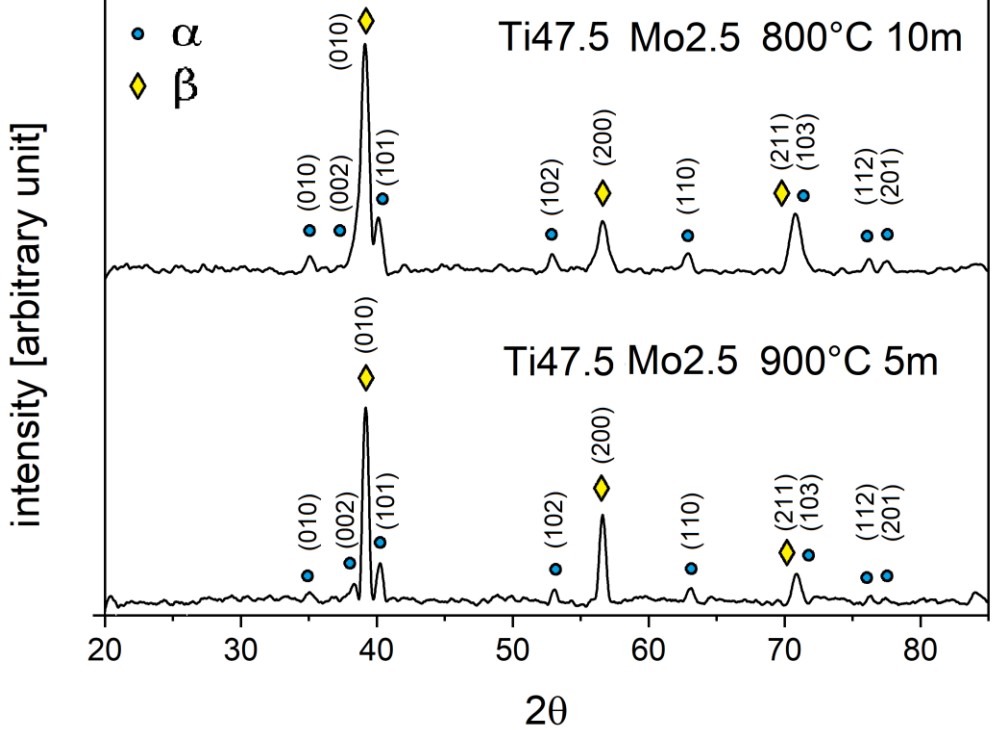

**Figure 1.** X-ray diffraction patterns of the porous TiMo alloy obtained by dealloying at 800 °C for 10 min (upper diagram) and 900 °C for 5 min (lower diagram).

Consistent with the XRD analysis, the microstructural analysis reveals two distinguished microstructural morphologies in the porous samples namely, dendrites and, so-called, ligaments (Figure 2). The ligaments are located in the interdendritic space and are connected with the dendrites. Such a complex morphology was reported for the open porous TiNb alloy [5] and TiFe [17] synthesized by liquid metal dealloying. The dendrites have been formed in the master $Ti_{47.5}Mo_{2.5}Cu_{50}$ alloy due to the limited solubility of Mo in Cu [45]. During solidification of the master alloy, the Ti-rich dendrites containing Mo solidify first and reject Cu-rich melt similarly to how it was observed in the recently developed titanium alloys [48–52]. Upon solidification, Cu-rich melt forms the matrix. Therefore, the dealloying of these Cu-rich matrix regions leads to the formation of porous structures in the interdendritic areas. This suggests the strategy to design dendrite-reinforced titanium alloys by liquid metal dealloying of TiCu-based precursors through the small addition of alloying elements (to the TiCu-based precursor) possessing limited solubility with Cu, e.g., Nb or Ta. Due to the complex morphology of the current TiMo porous samples, the ligament and dendrite sizes (thickness) were measured manually using ImageJ software. The ligament sizes of the samples dealloyed at 800 °C and 900 °C were found to be 0.75 ± 0.17 μm and 0.53 ± 0.15 μm, respectively. The thickness of dendrites from the samples dealloyed at 800 °C and 900 °C were found to be 0.94 ± 0.36 μm and 1.92 ± 0.62 μm, respectively.

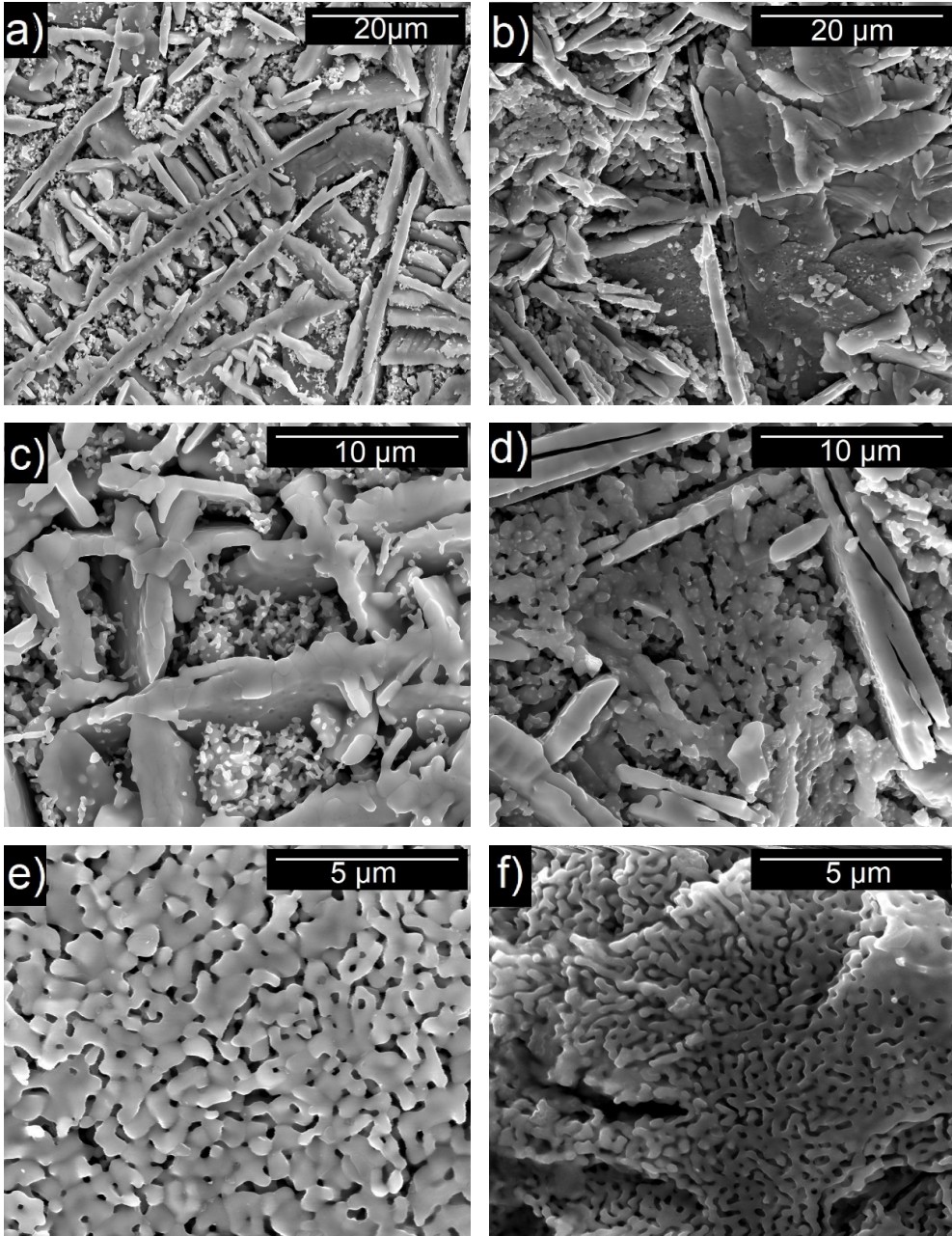

**Figure 2.** Scanning electron micrographs of the porous TiMo alloy obtained by dealloying at 900 °C for 5 min (**a,c,e**) and 800 °C for 10 min (**b,d,f**).

Figure 3 shows the EDX maps of the porous TiMo samples. Despite the immiscibility of Mo with Cu, Mo was found in the dendrites and porous areas of the samples. However, the concentration of Mo in the dendrites is higher when compared with that of the porous regions. This allows the assumption that the dendritic phase is likely to be the β-Ti phase and the porous phase is likely to be the α-Ti phase. The chemical composition of the phases is summarized in Table 1. Importantly for the biomedical application, the residual concentration of Cu is below 0.5 at.%. Generally, Cu is considered as a toxic element for human being [53]. However, low concentrations of Cu in an alloy can be beneficial for some biomedical applications due to the strong antibacterial effect [54,55]. The low Cu concentration also indicates that the liquid metal dealloying was completed.

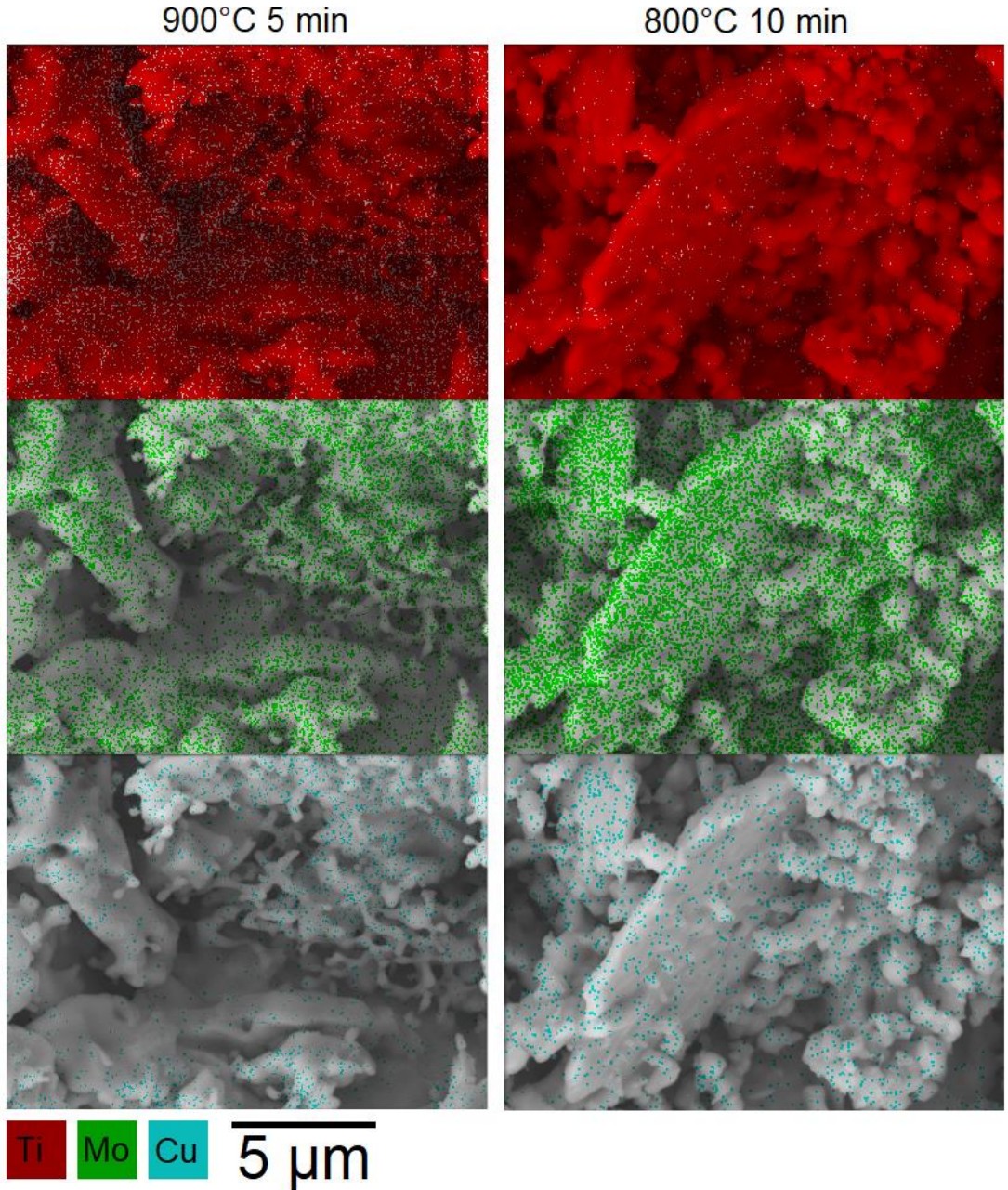

**Figure 3.** Elemental distribution maps of the porous TiMo alloy samples obtained by dealloying at 800 °C for 10 min (**right** panel) and 900 °C for 5 min (**left** panel).

**Table 1.** Chemical composition of the porous TiMo samples.

| Sample | Phase | Ti (at.%) | Mo (at.%) | Cu (at.%) |
|---|---|---|---|---|
| Sample 1 | Dendrite | 95.0 ± 0.02 | 4.6 ± 0.01 | 0.40 ± 0.01 |
|  | Porous | 96.0 ± 0.02 | 3.7 ± 0.01 | 0.30 ± 0.01 |
| Sample 2 | Dendrite | 94.7 ± 0.02 | 5.3 ± 0.01 | 0 |
|  | Porous | 97.1 ± 0.02 | 2.8 ± 0.01 | 0.10 ± 0.01 |

To get further insight into the complex microstructure of the current porous TiMo alloys, the samples were characterized using the X-ray nanotomography. Figure 4 shows the processed 3D tomographic images. The colors represent the thickness of the ligament. The yellow color corresponds to the thicker

ligaments and the blue to the thinner ligaments. The analysis of the ligament size based on the 3D images is shown in Figure 5. The ligament size distribution of the sample dealloyed at 800 °C can be well fitted by a Gaussian normal distribution (Figure 5a). The mean ligament thickness of this sample is 1.35 ± 0.38 µm. The sample dealloyed at 900 °C contains many data points possessing high width (Figure 5b), which probably occur from the dendrites, according to the SEM analysis. Unfortunately, the applied algorithm for thickness calculation can hardly differentiate between the ligaments and the dendrites. Therefore, it provides a large thickness mean value <D> of 2.49 ± 1.49 µm. To improve the evaluation of the mean ligament size and reduce the contribution of the dendrites, data points higher by a factor of 1.5 than the mean size have not been considered. In this case, the data points are well fitted with a normal distribution (blue curve) (Figure 5b). The distribution maximum is at 0.67 <D>, which results in a mean value of 1.66 ± 0.932 µm. The ligament size, calculated with a normal distribution from the reduced data, still leads to a high overestimation. Therefore, a log normal distribution was also applied (green curve). This follows the data better and results in a mean size of 0.42 <D> or 1.05 ± 1.7 µm. Typically, the thickness algorithm is known to overestimate the size of structural units of nanoporous metals by 30% [56]. Thus, the mean ligament size obtained from nanotomographical analysis is in a well agreement with the results obtained from the SEM analysis. Based on the tomography analysis, the pore size of the non-dendritic areas is 875 ± 690 nm for the sample dealloyed at 800 °C and 780 ± 490 nm for the sample dealloyed at 900 °C.

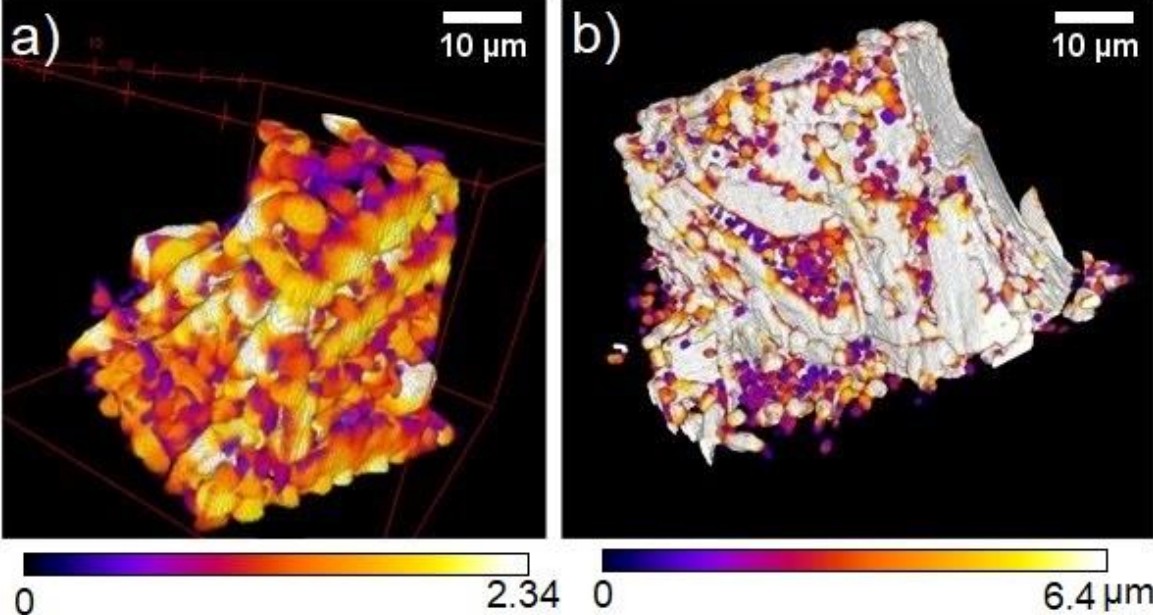

**Figure 4.** Colorized tomographic images of the open porous TiMo alloy obtained by dealloying at 800 °C for 10 min (**a**) and 900 °C for 5 min (**b**). Note: colors reflect thickness of ligaments; yellow corresponds to thicker and blue to thinner ligaments.

Despite the difference in microstructure, the samples fabricated using different parameters exhibit similar mechanical properties when probed under compression. The representative stress–strain curves are shown in Figure 6. The sample obtained by dealloying at a higher temperature exhibits a yield strength value of 180 ± 66 MPa while the sample dealloyed at 800 °C possesses a yield strength value of 172 ± 28 MPa (Table 2). These values are significantly higher than those found for the open porous α + β titanium TiNb alloy without dendrite-reinforcements [17] and are comparable with those found for the β titanium TiFe alloy containing dendrite-reinforcements [17]. The Young's modulus of both TiMo samples is below 10 GPa, which is in the range of human bone. Particularly, the Young's modulus of the sample dealloyed at 900 °C is 9.5 ± 1.1 GPa and for the sample dealloyed at 800 °C it is 8.2 ± 1.0 GPa. Both current samples possess a pronounced strain-hardening behavior, which

is beneficial for load bearing applications. Thus, considering the low values of Young's modulus and moderate strength in combination with the strain-hardening behavior, the current open porous materials reinforced by dendrites are promising for biomedical applications.

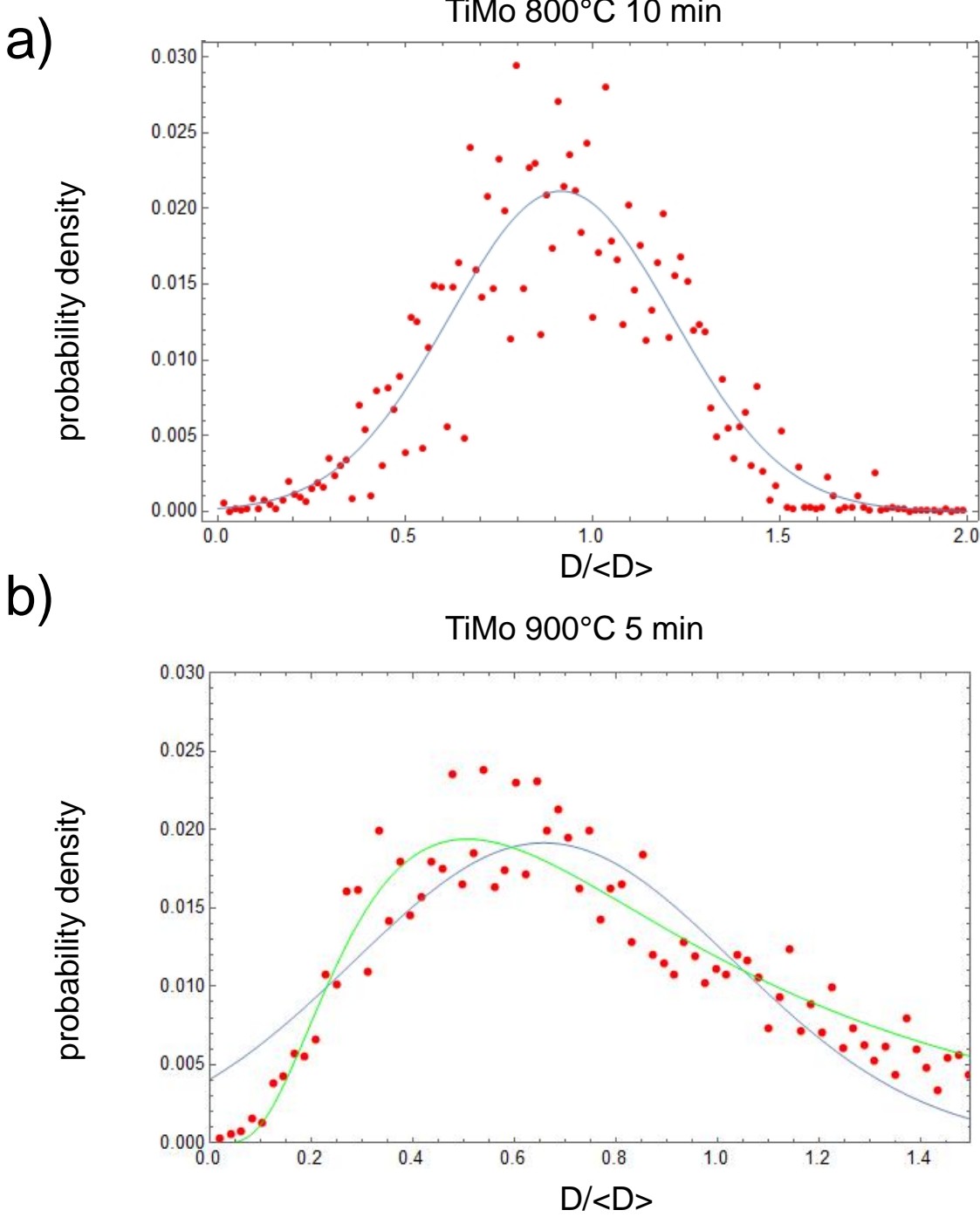

**Figure 5.** Ligament diameter distribution histograms of the open porous TiMo samples dealloyed at 800 °C for 10 min (**a**) and at 900 °C for 5 min (**b**). Note: Gaussian fit (blue) and log normal fit (green) of the scaled ligament width histograms scaled by the mean ligament width <D>.

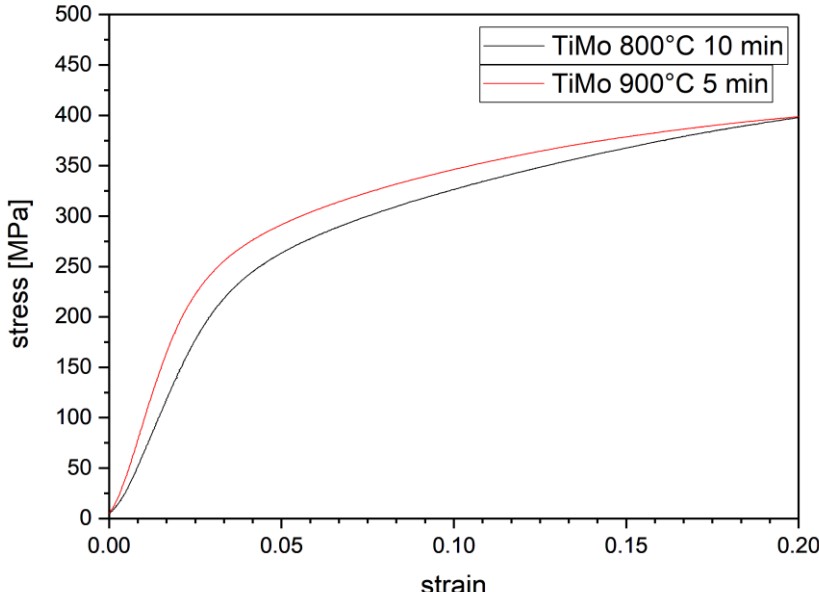

**Figure 6.** Mechanical behavior of the current open porous TiMo alloy. Representative compressive stress–strain curves of the porous TiMo alloy obtained by dealloying at 800 °C for 10 min and at 900 °C for 5 min.

**Table 2.** Mechanical properties with standard deviation of the porous TiMo samples.

| Sample | Yield Strength (MPa) | Young's Modulus (GPa) |
|---|---|---|
| TiMo 800 °C 10 min | 172 ± 28 | 8.2 ± 1.0 |
| TiMo 900 °C 5 min | 180 ± 66 | 9.5 ± 1.1 |

## 4. Summary

In this study, we have synthesized open porous α + β TiMo alloy by liquid metal dealloying of $Ti_{47.5}Mo_{2.5}Cu_{50}$ (at.%). The open porous titanium TiMo alloy possesses a complex microstructure and consists of micrometer scale dendrites surrounded by submicrometer scale ligaments. The dendrites are enriched in Mo and are composed of the bcc β-Ti phase while the ligaments are composed of the hcp α-Ti phase. Due to the limited solubility of Mo in Cu, the dendrites were already formed in the precursor $Ti_{47.5}Mo_{2.5}Cu_{50}$ alloy. Thus, the addition of alloying elements possessing limited solubility with Cu to the TiCu-based precursor promotes the formation of the dendrite-reinforced microstructure upon liquid metal dealloying. The current dendrite-reinforced porous TiMo samples possess the moderate strength of about 180 MPa and low Young's modulus below 10 GPa, as well as pronounced strain-hardening behavior. It was demonstrated that dealloying conditions influence the microstructure of the porous TiMo alloy while the mechanical behavior remains practically unchanged. The moderate strength, low elastic modulus and large deformability of the current open porous dendrite-reinforced TiMo alloy are favorable characteristics for biomedical applications.

**Author Contributions:** Conceptualization—I.V.O.; Formal analysis—S.A.B. and I.V.O.; Investigation—S.A.B.; Supervision—I.V.O.; Validation—S.A.B.; Writing – original draft, S.A.B. and I.V.O.; Writing-review & editing—S.A.B. and I.V.O. All authors have read and agreed to the published version of the manuscript.

**Funding:** This research received no external funding.

**Acknowledgments:** I.V.O. is grateful for the financial support provided by the German Science Foundation under the Leibniz Program (grant MA 3333/13-1). We also want to acknowledge the fruitful discussion with Jürgen Markmann.

**Conflicts of Interest:** The authors declare no conflict of interest.

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
