# Peer review of "Open Porous α + β Titanium Alloy by Liquid Metal Dealloying for Biomedical Applications"

_metals, doi:10.3390/met10111450_

Round 1
Reviewer 1 Report
The manuscript by Berger and Okulov titled “Open porous α + β titanium alloy by liquid metal dealloying for biomedical applications”. The topic is very interesting, however, the manuscript contains some issues that should be addressed before publication. Thus, I suggest this work to be accepted with major revisions.
Some comments to be addressed:
General comments:
- The authors titled the manuscript as “open porous “, however through all the manuscript it is not possible to observe the porosity. Thus, some questions may be arise: Which is the pores sizes and morphology. Which is the porosity level? Which is the percentage of close and open pores?
- Some typing mistakes can be found in the manuscript such as:
- 2 line 67 – HNO3, 3 should be subscript
- Figure captions of figures 1, 4, 5, and 6 symbol of temperature is not correct.
Introduction:
- It is not clear which is the target application
Experimental Materials and Methods
The different processing conditions should be given, e.g. 800 °C for 10 min.
Results and discussion
This section is a description of the obtained results. However, there is a lack of discussion of these results with the literature.
- 2 line 78/79 – “(…) two different conditions”, which conditions?
- 3 line 84/85 – Are these microstructures over-etched? Why the authors did not showed the microstrcture without being over-eteched?
- 4 line 111/112 – “The low Cu concentration also indicates that the liquid metal dealloying was completed”. Do the authors know the Cu concentration?
Author Response
Thank you for all the helpful comments. Find our reply attached

Reviewer 2 Report
The paper presents a study on liquid metal dealloying (LMD) of a bi-phased eutectic Ti alloy in order to generate an architectured nano-porous material. The precursor is chosen in a way that only one phase will dealloyed in liquid Mg. The resulting material presents simultaneously good mechanical properties because of the non-dealloyed phase and classical open nano-porous structure of liquid metal dealloyed samples.
The originality of this study is to successfully apply the LMD process to a bi-phased material to dealloyed only one phase.
The authors show that the resulting materials exhibit mechanical behavior comparable and compatible with human bones, with open nano-porous foam which opens promising biomedical application.
The paper deserves to be publish, after some improvements: I have 2 main recommendations.
Best regards
1/ most of the shown results suffer from the difficulty to analyze nano-foam, especially for multi-scale architectured material. However, despite the frequent reference to the eutectic Ti precursor, there is no study of it. The paper would greatly be improved by adding the comparison of the result with the same done on the precursor. As a bulk material, precursor samples shouldn't suffer the same difficulty of preparation. The EDX analysis will confirm the nature of the 2 eutectic phases, SEM pictures will help with the initial morphology and the phase quantification and composition, and tomography should even be possible considering the large Z-contrast between Cu and Ti atoms given access to the full 3D-microstructure ; all before dealloyiing of course.
2/ Figure 6 is missing a bit of preparation/presentation. Why are there 2 curves per graph ? why are they different ? Is there a meaning for the color ?
As the authors compare them with bones behavior, would it be possible to add a bone stress/strain curve ?
Author Response
Thank you for the comments. Find our reply attached

Reviewer 3 Report
This is an interesting study but lacks penetration on the metallurgical aspects. The alloy system is complex but could be defined well by using appropriate thermochemical software such as Thermocalc. It would be very useful to the reader to follow the changes using an equilibrium phase analysis which could also explain the final compositions achieved. As a comment it would also be useful to indicate why this method is medically useful as opposed to for example making the same structure by using powder consolidation techniques.
Author Response

(The authors gave the same response as above.)

Reviewer 4 Report
The manuscript reports on the synthesis of open porous α + β titanium alloy by liquid metal dealloying. The presented results would be interesting to materials communities. Thus, the manuscript can be recommended to be published in Metals after the authors addressing the following comments/concerns.
Methods
Line 62 – In the section "Methods" it was not specified that different dealloying conditions were used, and also what caused the choice of just such conditions.
Line 69 – How was the identification of the reflections on the XRD spectra carried out?
Line 74 – It is not entirely clear from the text of the article what exactly were the dimensions of the samples for mechanical tests and how they were prepared.
Results and Discussions
Line 81 – Please label the lattice planes on each peak on XRD spectra.
Line 120 – within what limits does the thickness of the ligament change? Specify the range for blue and yellow colours in Fig.4.
Lines 145-156 – The reviewer believes that the mechanical properties of samples fabricated using different parameters should also be presented in a table for better understanding and easy comparison.
Line 157 – Figure 6 - captions a and b are not used in any way in the text. It is not indicated what curves are shown in the figure (red and black curves, figure 6a; blue and black curves, figure 6b).
However, these comments do not reduce the relevance and importance of the results. The article is sufficiently novel and interesting to warrant publication. Obtained data are sufficient to reproduce the experiment. Results and discussion are reliable. References reflect the main publications on which the work is based.
The reviewer believes that the manuscript needs revisions of indicated concerns in order to be published.
Author Response

(The authors gave the same response as above.)

Round 2
Reviewer 2 Report
typo line 145
line 146 : size of 780+-490nm : I guess 780 is 1050-30%, but I can't understand where does the 490 comes from...
What is the physical meaning of the lognormal distribution in this case ?
it would have also been interesting to look at the porosity size, tortuosity, etc...
About the authors answers :
Response 1: We agree that investigation of the precursor might be of
interest, but it will not change the main message of the study presented which is synthesis of open porous materials possessing similar elastic behavior as human bone.
=> I strongly disagree. The mechanical properties are due to the use of the perlitic structure of the precursor. Characterization of this precursor is of utmost importance. Else, the authors have shown that it is possible to have an open porous material possessing similar elastic behavior as human bone, starting with a semi-mysterious precursor.
Is it reproductible ? how ?
I am quite aware of what is possible with tomography. If the perlitic structure is micrometric, you don't need nano-tomo to catch it, and that would give you the needed information.
Response 2 :
=> OK for the curves and the table, but : what is the meaning of the +/- uncertainties in the table ? how many sample where analyzed ?
If only 2, then just give the 2 values for each property. Average and standard deviation for 2 values is just meaningless...
If more, why only 2 curves ? how did you chose them ?
Round 3
Reviewer 2 Report
The study and more generally the domain of LMD would deserved much more time and people involved I think, considering how interesting the subject is (both on fundamental and applied point of view).
I understand very well that one can't publish a full thesis on just a paper, therefore I recommend to accept the publication as it is.
I thank the authors for the exchange, I do learn things while reading the paper and there comments. And I hope they will continue to work on LMD.
Thank you for trusted me as reviewer.
I wish you the best, especially on this special time.
Best regards